# Clinical utility of peptide microarrays in the serodiagnosis of neglected tropical diseases in sub-Saharan Africa: protocol for a diagnostic test accuracy systematic review

Arthur Vengesai  ,[1,2] Thajasvarie Naicker,[2] Maritha Kasambala,[3] Herald Midzi,[1,2] Tariro Mduluza-Jokonya  ,[2] Simbarashe Rusakaniko,[4] Takafira Mduluza[1]

¹Biotechnology and Biochemistry, University of Zimbabwe Faculty of Science, Harare, Zimbabwe
²College of Health Sciences, University of KwaZulu-Natal, Durban, South Africa
³Faculty of Science and Agriculture, University of KwaZulu-Natal, Durban, South Africa
⁴College of Health Sciences, University of Zimbabwe, Harare, Zimbabwe

**Correspondence to**
Mr Arthur Vengesai;
arthurvengesai@gmail.com

## ABSTRACT

**Introduction** Neglected tropical diseases tend to cluster in the same poor populations and, to make progress with their control, they will have to be dealt with in an integrated manner. Peptide microarrays may be a solution to these problems, where diagnosis for co-infection can be detected simultaneously using the one tool. A meta-analysis using hierarchical models will be performed to assess the diagnostic accuracy of peptide microarrays for detecting schistosomiasis (*Schistosoma mansoni* and *S. haematobium*), soil-transmitted helminths (*Trichuris trichiura*, *Ascaris lumbricoides* and *Necator americanus*), trachoma (*Chlamydia trachomatis*), lymphatic filariasis (*Wuchereria bancrofti*) and onchocerciasis (*Onchocerca volvulus*) in people residing in sub-Saharan Africa.

**Methods and analysis** A comprehensive search of the following databases will be performed: Cochrane Infectious Diseases Group Specialised Register, PubMed, EMBASE and The Web of Science. Studies comparing peptide microarrays with a reference standard from a random or consecutive series of patients will be included in the study. Two review authors will independently screen titles and abstracts for relevance, assess full-text articles for inclusion and carry out data extraction using a tailored data extraction form. The quality Assessment of Diagnostic Accuracy Studies-2 tool will be used to assess the quality of the selected studies. The bivariate model and the hierarchical summary receiver operating characteristic curve model will be performed to evaluate the diagnostic accuracy of the peptide microarrays. Meta-regression analyses will be performed to investigate heterogeneity across studies.

**Ethics and dissemination** There is no requirement for ethical approval because the work will be carried out using previously published data, without human beings involvement. Findings will be disseminated through peer-reviewed publication and in conference presentations.

**PROSPERO registration number** CRD42020175145.

### Strengths and limitations of this study

► This systematic review protocol follows the Preferred Reporting Items for Systematic Review and Meta-Analysis Protocols guidelines.
► This systematic and meta-analysis review will be the first to explore the diagnostic test accuracy of peptide microarrays.
► The methods described in the review may be applicable to any diagnostics test accuracy systematic and meta-analysis review.
► There is the potential for poor quality in the reporting of diagnostics test accuracy studies.
► Excluding articles not published in English may lead to publication bias.

40% of whom live on the African continent.[1] Notably, NTDs affect the world's poorest, most marginalised and remote communities, where access to clean water, sanitation and healthcare is limited. The impact of NTDs on communities are devastating; they cause severe pain, disabilities, deformities, malnutrition, stunted growth, cognitive impairment, social isolation and humiliation. These diseases may be fatal, in fact anaemia caused by some NTDs have a direct impact on maternal mortality. Importantly, NTDs have a disruptive impact on the productivity of already unstable economies. They keep children out of school and adults out of work, hence trapping poor communities in endless cycles of poverty.[1–3]

Sub-Saharan Africa is estimated to account for the following worldwide NTD proportions: approximately 25%–33% of soil-transmitted helminth (STH) infections, more than 33% of lymphatic filariasis infections, 50% of trachoma infections, most of the world's cases of schistosomiasis, human

## INTRODUCTION

Neglected tropical diseases (NTDs) are a group of debilitating communicable diseases that affect over 1.6 billion people globally,

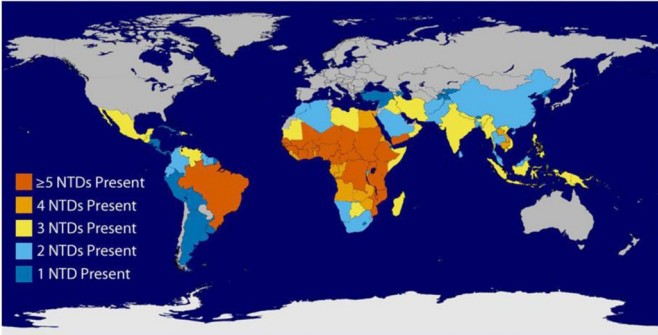

**Figure 1** Global overlap of the six most common neglected tropical diseases (NTDs).[44]

African trypanosomiasis, loiasis and onchocerciasis, and all of the world's cases of dracunculiasis and human African trypanosomiasis.[4 5] Different NTDs can occur in the same poor populations[6] (illustrated in figure 1). In most regions of sub-Saharan Africa, it is not uncommon to find five or more NTDs in one area—the three major soil-transmitted helminth infections, schistosomiasis, lymphatic filariasis, onchocerciasis and trachoma.[7] Zimbabwe, a country in Southern Africa, is endemic to four of the most common NTDs, lymphatic filariasis, schistosomiasis, STH and trachoma.[8–10]

The lack of rapid, accurate, simple-to-use, point-of-care tests for many of the neglected tropical diseases is an important feature for their general neglect and the under-appreciation of their disease burden.

Diagnosis plays a crucial role in the surveillance and detection of infectious diseases including NTDs.[11] However, NTDs remain mostly undiagnosed, the reliable identification of parasitic infections requires in-depth training for specimen preparation, and expertise for subsequent microscopic examination are unavailable in most rural clinics and remote areas.[12 13] The diagnosis of NTDs is primarily based on well-established and widely used laboratory techniques, such as the examination of blood, stool and urine samples under a microscope.[12] Schistosomiasis is diagnosed through the microscopic detection of parasite eggs in stool or urine specimens using the Kato-Katz and urine filtration techniques, respectively.[14] The standard method for diagnosing the presence of STHs is by microscopically identifying STH eggs in a stool sample using the Kato-Katz technique or the formal ether concentration technique.[15] The standard method for diagnosing lymphatic filariasis active infection is the identification of microfilariae in a blood smear by microscopy.[16] Trachoma diagnosis relies on the use of conjunctival examination for the clinical sign of trachomatous inflammation by a trained eye specialist. PCR-based assays for trachoma diagnosis are technically complex and too expensive.[17] Microfilariae microscopic examination of skin biopsies (snips) is the gold standard test for diagnosis of onchocerciasis. The biopsy is performed using a sclerocorneal biopsy punch or by elevating a small cone of skin with a needle and slicing it off with a scalpel to obtain bloodless skin.[18 19]

While microscopy of parasites is considered as highly specific, its sensitivity depends on the intensity of infection and timing of sample collection. Schistosomiasis eggs tend to be passed at irregular intervals and in small amounts and may not be detected.[20] STH eggs may be difficult to find in light infections,[15] hence concentration techniques are often recommended for diagnosis. The microfilariae that cause lymphatic filariasis circulate in the blood at night and blood collection should be done at night to coincide with the appearance of the parasite.[16] For trachoma, conjunctival inflammation may persist in the absence of detectable bacteria, an important limitation to decision-making based on clinical examinations.[17] Furthermore, in Loiasis endemic areas, the specificity of skin snips for the diagnosis of onchocerciasis is limited. Performing PCR of the skin snip can increase the sensitivity and specificity; however, the PCR tests are not commercially available.[18 21]

Serological tests provide an alternative to microscopic detection of parasites for the diagnosis of schistomiaisis, STHs, trachoma, lymphatic filariasis and onchocerciasis. Antibodies and/or antigens detected in blood samples are also indications of *Schistosoma* infections.[22] Patients with active filarial infection typically have elevated levels of anti-filarial IgG4 in the blood, and these can be detected using serological assays.[16] A dipstick immunoassay based on detection of chlamydial lipopolysaccharide was developed for diagnosis of trachoma.[17] Serological tests may also be useful in showing exposure to infection and the need for thorough examination, treatment and follow-up in people living in low-transmission areas.[22]

Engels and Savioli[6] recommended diagnosis of NTDs collectively as a group, as they tend to cluster in the same poor populations and, to make progress with their control, they need to be dealt with in an integrated manner. Peptide microarrays may be a solution to these problems, where NTD co-infections can be diagnosed simultaneously using one sample and tool.

Peptide microarrays (collections of short peptides of pathogens immobilised on solid planar supports) are large-scale screening systems for simultaneous identification of multiple pathogens from small quantities of serum or plasma and other body fluids.[23] The principle behind peptide microarrays involves the use of antibody tests that can determine whether an individual is infected or uninfected. Antibody–protein interactions play a critical role in the humoral immune response. B cells secrete antibodies, which bind antigens. The specific part of antigens that are recognised by antibodies are called B-cell epitopes. These epitopes may be short linear (continuous) peptides, corresponding to contiguous amino acid residues on the surface of an antigen, or conformational (discontinuous), in which case the residues are not sequential in the primary structure, but are in close proximity within the folded protein 3D structure.[24 25]

The Expanded Special Project for Elimination of Neglected Tropical Diseases (ESPEN) works to reduce the burden of five targeted NTDs, schistosomiasis, STHs,

trachoma, LF and onchocerciasis, which put at an estimated 600 million people at risk in Africa.[3] Against this background, the objective of the review is to assess the diagnostic accuracy of peptide microarrays for the detection of schistosomiasis (*Schistosoma mansoni, S. haematobium*), STHs (*Trichuris trichiura, Ascaris lumbricoides* and *Necator americanus*), lymphatic filariasis (*Wuchereria bancrofti*), trachoma (*Chlamydia trachomatis*) and onchocerciasis (*Onchocerca volvulus*) in people living in endemic areas. Diagnostic accuracy will be determined by sensitivity and specificity of each test.

## METHODS

Our methodology for the study will be based, on guidelines proposed by the Cochrane Handbook for Systematic Reviews of Diagnostic Accuracy and from relevant literature.[26–28] The study will also use the Preferred Reporting Items for Systematic Review and Meta-Analysis (PRISMA) guidelines published by Moher *et al* to search and select the studies to be included in the systematic review.[29] The PRISMA checklist (online supplemental file 1) will be used to ensure the inclusion of relevant information in the analysis and the quality assurance appraisal will be used as the basis for literature selection.

### Search strategy
#### Electronic searches
A comprehensive search of the literature will done in the general databases PubMed, EMBASE, Web of Science and Cochrane Infectious Diseases Group Specialised Register. The search will be between the inception of each database to present day, using the search terms and Boolean operators shown in tables 1–5. Alternative terms and concepts which address the same question will be used as it is common for a range of terms to be used to describe the same phenomenon or research area. Figure 2 shows the initial search for *S. haematobium* in PubMed. After removing duplicates, articles will be assessed for inclusion into the review through title and abstract analysis. Shortlisted articles will be screened for inclusion through full-text analysis.

### Searching other resources
We will search the proceedings and abstracts of relevant conferences. We will hand search the reference lists of all the relevant primary studies for additional relevant publications for inclusion in the study. We will also search for relevant articles using the PubMed "similar articles" function. PEPperPRINT Heidelberg Germany and other companies involved in the manufacture of peptide microarrays will be contacted to obtain additional unpublished literature from their collaborators.

### Criteria for considering studies for this review
#### Types of studies
Studies evaluating diagnostic test accuracy of peptide microarrays in a consecutive series of patients, or a randomly selected series of patients, will be eligible. Where the report does not explicitly state that sampling was consecutive, but consecutive sampling was judged most probable, the study will be included. Studies will be excluded if they do not present sufficient data to allow us to (1) extract or calculate absolute numbers of true positives, false positives, false negatives and true negatives and (2) they provided insufficient information to calculate sensitivity and specificity. Studies will also be excluded if they were not available in English or if they presented insufficient information to fully assess their eligibility. Review articles, studies of retrospective design where investigators collected samples after execution of the reference test, abstracts of meetings and case reports where full text is not available will be excluded from the study.

**Table 1** Search strategy for schistosomiasis

| Microarray OR Biochip OR Microchip 0R Micro chip OR Microarray OR Chip OR Array | AND | Epitope OR Peptide | AND | Schistosomiasis OR Bilharzia OR *S. haematobium* OR *S. mansoni* OR *Schistosoma haematobium* OR *Schistosoma mansoni* |
|---|---|---|---|---|

**Table 2** Search strategy for soil transmitted helminths

| Microarray OR Biochip OR Microchip 0R Micro chip OR Microarray OR Chip OR Array | AND | Epitope OR Peptide | AND | Roundworms OR hookworm OR whipworm OR *A. lumbricoides* OR *Nector americanus* OR *Trichuris trichiura* |
|---|---|---|---|---|

**Table 3** Search strategy for trachoma

| Microarray OR Biochip OR Microchip 0R Micro chip OR Microarray OR Chip OR Array | AND | Epitope OR Peptide | AND | Trachoma OR Blinding trachoma OR Granular conjunctivitis OR Egyptian opthalmia OR *Chlamydia trachomatis* |
|---|---|---|---|---|

| Table 4 | Search strategy for lymphatic filariasis | | | |
|---|---|---|---|---|
| Microarray OR Biochip OR Microchip 0R Micro chip OR Microarray OR Chip OR Array | AND | Epitope OR Peptide | AND | Lymphatic filariasis OR elephantiasis OR *Wuchereria bancrofti* |

## Participants

Studies recruiting individuals from sub-Saharan Africa will be included in the study. Studies will be excluded if participants had been treated for the target conditions and the tests would have been performed to assess the treatment outcomes. In studies with broader inclusion criteria but which presented results stratified by subgroups, we will include the data relevant to our inclusion criteria.

## Index tests

Studies evaluating all peptide microarray diagnostic tools, designed to detect the target conditions in parallel (multiplex diagnostics), or individually will be included in the study. In addition, studies that exclusively used full-length proteins for microarray immunoassays will also be included in study

## Comparator tests

Studies will be included regardless of whether they made comparisons with other microarray or serological tests.

## Target conditions

Studies aimed to detect schistosomiasis (*S. mansoni, S. haematobium*), STHs (*Trichuris trichiura, Ascaris lumbricoides* and *Necator americanus*), trachoma (*Chlamydia trachomatis*), lymphatic filariasis (*Wuchereria bancrofti*) and onchocerciasis (*Onchocerca volvulus*) either in parallel or individually will be included in this review article.

## Reference standards

Studies that would have used the gold standard diagnostic method of each disease as the reference standard (table 6) will be included in the study.

| Table 5 | Search strategy for onchocerciasis | | | |
|---|---|---|---|---|
| Microarray OR Biochip OR Microchip 0R Micro chip OR Microarray OR Chip OR Array | AND | Epitope OR Peptide | AND | Onchocerciasis OR River blindness OR *Onchocerca volvulus* |

## Data collection and analysis

### Selection of studies

Two authors will independently assess the titles of studies identified by the search, excluding those obviously irrelevant to the diagnosis of the target conditions using peptide microarrays. A single failed eligibility criterion will be sufficient for a study to be excluded from the review study. Letters, review articles, and articles clearly irrelevant based on examination of the abstract and other notes will be excluded next and the eligibility of the remaining potentially relevant articles will be judged on full-text publications.

### Data extraction (selection and coding)

Campbell *et al*,[30] using standards for the reporting of diagnostic accuracy studies (STARD) checklist, developed the data extraction tool that will be used in this review. The following data will be extracted.

► Study authors, publication year, and journal.
► Study design.
► Case definition.
► Study participants—age, sex.
► Prevalence of target condition.
► Reference standard (including criteria for positive test).
► Index tests (cut-off values (pre-specified or not) and whether the test was a commercial of in-house test).
► Geographical location of data collection.
► Index/reference time interval (treatment status of participants before study or post treatment).
► Distribution of severity of disease in those with target condition
► Other diagnoses in those without target condition.
► Infection intensity (egg counts in urine and stool by microscopy).
► Presence of missing or unavailable test results.
► Numbers of true positives, true negatives, false positives, and false negatives.

Data extraction will be conducted independently by two authors to avoid bias. Any discrepancy will resolved by discussion. Where an agreement cannot be reached, a third author will be consulted. Where it remains unclear whether a study is eligible for inclusion, it will be excluded.

For each comparison of the index test with the reference test, data will be extracted on the number of true positives, true negatives, false positives and false negatives in the form of a two-by-two table. Like all studies of diagnostic test accuracy that comply with the STARD statement, the two-by-two tables will further be used to calculate sensitivities and specificities of the target assays. Where two versions of one reference standard or index test are used, for example, local clinic and expert standard microscopy or field versus laboratory testing, only the one most likely to yield the highest quality results will be included in the review. In cases of studies where only a subgroup of participants met the review inclusion criteria, data will be extracted and presented only for that particular subgroup.

| Search | Add to builder | Query | Items found | Time |
|---|---|---|---|---|
| #16 | Add | Search ((((((Microarray Analysis[MeSH Terms]) OR Microarrays) OR Microarray[Title/Abstract]) OR Array[Title/Abstract])) AND ((((Peptides[MeSH Terms]) OR Epitopes[MeSH Terms]) OR Peptide[Title/Abstract]) OR Epitopes[Title/Abstract])) AND ((((Urogenital schistosomiasis) OR Schistosoma haematobium) OR S. haematobium) OR schistosomiasis) | 56 | 04:53:56 |
| #15 | Add | Search (((Urogenital schistosomiasis) OR Schistosoma haematobium) OR S. haematobium) OR schistosomiasis | 27604 | 04:53:32 |
| #14 | Add | Search schistosomiasis | 27038 | 04:53:06 |
| #13 | Add | Search S. haematobium | 2547 | 04:52:51 |
| #12 | Add | Search Schistosoma haematobium | 3773 | 04:52:37 |
| #11 | Add | Search Urogenital schistosomiasis | 2548 | 04:51:59 |
| #10 | Add | Search (((Peptides[MeSH Terms]) OR Epitopes[MeSH Terms]) OR Peptide[Title/Abstract]) OR Epitopes[Title/Abstract] | 2902261 | 04:51:11 |
| #9 | Add | Search Epitopes[Title/Abstract] | 49899 | 04:50:47 |
| #8 | Add | Search Peptide[Title/Abstract] | 406530 | 04:50:16 |
| #7 | Add | Search Epitopes[MeSH Terms] | 111708 | 04:49:55 |
| #6 | Add | Search Peptides[MeSH Terms] | 2672990 | 04:49:28 |
| #5 | Add | Search (((Microarray Analysis[MeSH Terms]) OR Microarrays) OR Microarray[Title/Abstract]) OR Array[Title/Abstract] | 275317 | 04:48:52 |
| #4 | Add | Search Array[Title/Abstract] | 146900 | 04:46:23 |
| #3 | Add | Search Microarray[Title/Abstract] | 87844 | 04:45:54 |
| #2 | Add | Search Microarrays | 30663 | 04:45:28 |
| #1 | Add | Search Microarray Analysis[MeSH Terms] | 91509 | 04:45:05 |

**Figure 2** Search strategy for *S. haematobium* in PubMed.

## Quality of the studies (risk of bias assessment)

In order to reduce bias, two reviewers will independently assess the quality of each individual study. In cases of disagreements, a third reviewer will be consulted in order to resolve the disagreement. The most widely used tool for examining diagnostic test accuracy QUADAS-2 tool (Quality Assessment of Diagnostic Accuracy Studies-2) will be used to assess the quality of studies that will be included. Critical appraisal questions on the checklist will be answered with a yes/no response or will be marked unclear if insufficient information was reported to allow judgement to be made.[30 31]

## Statistical analysis

We will stratify all diagnostic tests by the target condition and reference standards used. For each study within each stratum, we will construct two-by-two tables for true positives, true negatives, false positives and false negatives. We will construct forest plots displaying sensitivity and specificity of the index test from the contingency tables assuming that the reference methods will be 100% sensitive and specific. Where only sensitivity and specificity estimates are reported, we will derive the two-by-two table from the reported data. We will enter the two-by-two data into Review Manager (RevMan) software for Windows (Cochrane Collaboration, Copenhagen, Denmark). Estimates of sensitivity and specificity from each individual study will be summarised on forest plots at 95% CI and plotted using summary receiver operating characteristic (SROC) plots.

Meta-analysis will be performed in cases where three or more studies are available for the index test and target condition. Studies will also be submitted to meta-analysis when three conditions are met: sample size is greater than 20; sensitivity and specificity are available for the index; when a control group is included in the analysis. Hierarchical/multi-level random models, including hierarchical summary receiver operating characteristic (HSROC) model and the bivariate random-effects model, will be used in our study to evaluate the diagnostic accuracy of peptide microarrays. These models allows for both within-study and between-study heterogeneity.[32] The bivariate model models sensitivity and specificity directly at a common threshold, and its primary objective is to obtain a summary estimate of sensitivity and specificity. It is recommended for purely binary test or when different studies report similar thresholds.[33–35] HSROC includes a random-effects term for variation in accuracy and threshold between studies, and non-symmetrical underlying ROC curves. This model estimates the underlying ROC curve, which describes how sensitivity and specificity of the included studies trade off with each other as thresholds vary. The position of the ROC curve depends on the degree of overlap of the distributions of the diseased and non-diseased and helps to estimate the level of discriminatory power of a test. The closer the ROC curve is to

| Table 6 | Selected NTDs and their gold standard diagnostic tests |
|---|---|
| **Disease/pathogen** | **Reference standard** |
| *S. haematobium* | Microscopic detection of parasite eggs present in urine using the urine filtration[45] |
| *S. mansoni* | Microscopic detection of parasite eggs present in stool using the Kato-Katz techniques. Formal ether concentration method and the point-of-care circulating cathodic antigen (POC-CCA) will be considered as alternative reference standards[45] |
| STHs (*Trichuris trichiura*, *Ascaris lumbricoides* and *Necator americanus*) | Microscopic detection of parasite eggs present in stool using the Kato-Katz techniques. The formal ether concentration technique will be considered as an alternative reference standard[45] |
| *Chlamydia trachomatis* | Blinding will be diagnosed clinically using the WHO grading system for trachoma[46] |
| *Wucheria bancrofti* | Microscopic examination of microfilariae in a blood smear[16] |
| *Onchocerca volvulus* | Skin snip biopsy[18] |

NTD, neglected tropical disease.

the upper-left corner of the graph, the better the tests discriminate between diseased and non-diseased. In general, the HSROC model is recommended for continuous tests when the included studies all report a different threshold for test positivity will be used if data from a test has multiple thresholds.[32 34]

In this review, we will use the bivariate model when data from a test had one or a common threshold to perform the overall meta-analysis. In scenarios where there is between-study variation in thresholds, we will perform meta-analyses by using the HSROC model to estimate SROC curves of sensitivity and specificity. The summary estimates of sensitivity and specificity and 95% CI and the HSROC will calculated using MetaDTA (an interactive web-based tool to conduct and interrogate meta-analysis of diagnostic test accuracy studies).[36]

### Assessment of publication bias
In this review, Begg's rank correlation test[37] and Egger's regression test[38] will be used to evaluate possible publication bias. Publication bias will be assessed when five or more studies are available.

### Investigations of heterogeneity
We will visually inspect forest plots and summary ROC plots to investigate the heterogeneity between study-specific estimates of sensitivity and specificity. When five or more studies are available, we will perform meta-regression analysis to explore the heterogeneity between studies. We will investigate heterogeneity by adding covariates (including, age group, gender, infection intensity and country) in the bivariate model which allows for separate calculations of sensitivity and specificity and of the effect of each of the covariates on summary results of sensitivity and specificity. Meta-regression analysis will be performed in OpenMeta-Analyst for Windows 10 (open-source, cross-platform for advanced meta-analysis).

### Patient and public involvement
There will be no patient or public involvement in conducting this review article because it will be based on previously published data.Ethics and dissemination

### DISCUSSION
Serological assays commonly used for the diagnosis of infectious diseases such as dengue infection are relatively inexpensive and easy to perform compared with biochemical tests, cultures or nucleic acid–based methods.[39] In cases where faecal specimens are unavailable, serology can have a role in diagnosis of STHs and *S. mansoni*.[13] However, for serodiagnosis of human helminthic infections, many tests in use rely on native antigens, either extracted from whole pathogens maintained in laboratory animals or cultivated in vitro to obtain metabolic antigens. These natural antigens are limited in availability and suffer from batch-to-batch variation, and their production is laborious. Recombinant antigens

used in serodiagnostic tests require a high degree of purification to avoid cross-reactivity due to contaminants from the expression system.[40] The limitations associated with native antigens and recombinant antigens, such as unspecific binding and cross-reactivity in serological diagnosis of various diseases, may be resolved by the use of standardised and highly pure synthetic peptide.[40] Studies on leishmaniasis, Chagas disease, schistosomiasis, paracoccidioidomycosis, tuberculosis and, more recently, on cryptococcosis, among others, have shown that this approach has potential for the early diagnosis of disease, thus reducing the morbi-lethality of individuals affected by these infections and ultimately changing their prognosis.[41 42]

For improving diagnostic test performance, it is desirable to identify highly specific and highly reactive epitopes from the proteome of the pathogen in question and synthetically produce the corresponding peptide antigens. Synthetic peptides are advantageous for diagnostic applications since they are well defined, easily produced in large amounts, highly pure and often cost-saving if compared with the production of natural antigen in animal models or in vitro culture.[40]

Also in light of the fact that many cases of infectious diseases are still under-reported or misclassified, peptide microarray analysis opens up new perspectives for the use of antibodies as diagnostic biomarkers for disease surveillance and provides unique access to a more differentiated serological diagnosis.[43] Hence, in the present study, we will conduct a systematic review that focuses on the latest applications of peptide microarrays in the serologic diagnosis and surveillance of NTDs.

**Acknowledgements** The authors would like to acknowledge the valuable input of Professor Francisca Mutapi.

**Contributors** AV and TM conceived and designed the study protocol. AV, HM, TM-J and MK will conduct a comprehensive literature search, screen the literature and extract data. AV will write the review paper. TN, SR and TM will provide guidance. All authors will read and approve the final version of the manuscript. All authors have read and approved this protocol.

**Funding** This research was commissioned by the National Institute for Health Research (NIHR) Global Health Research programme (16/136/33) using UK aid from the UK Government. This publication was also supported by a grant from the United States Agency for International Development (USAID) and UK aid from the British people (UK aid) through the Coalition for Operational Research on Neglected Tropical Diseases (COR-NTD) and administered by the African Research Network for Neglected Tropical Diseases (ARNTD). Grant number SGPIII/0210/238.

**Disclaimer** The views expressed in this publication are those of the authors and not necessarily those of the NIHR or the Department of Health and Social Care. The contents are solely the responsibility of the authors and do not necessarily represent the views of USAID, UK aid, COR-NTD or the ARNTD.

**Map disclaimer** The inclusion of any map (including the depiction of any boundaries therein), or of any geographic or locational reference, does not imply the expression of any opinion whatsoever on the part of BMJ concerning the legal status of any country, territory, jurisdiction or area or of its authorities. Any such expression remains solely that of the relevant source and is not endorsed by BMJ. Maps are provided without any warranty of any kind, either express or implied.

**Competing interests** None declared.

**Patient and public involvement** Patients and/or the public were not involved in the design, or conduct, or reporting, or dissemination plans of this research.

**Patient consent for publication** Not required.

 Vengesai A, *et al. BMJ Open* 2021;**11**:e042279. doi:10.1136/bmjopen-2020-042279

**Provenance and peer review** Not commissioned; externally peer reviewed.

**ORCID iDs**
Arthur Vengesai http://orcid.org/0000-0002-7629-6843
Tariro Mduluza-Jokonya http://orcid.org/0000-0002-1219-6464

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
