## [Reviewer comments · BMJ Open]

ARTICLE DETAILS

TITLE (PROVISIONAL)	Clinical utility of peptide microarrays in the serodiagnosis of neglected tropical diseases in sub-Saharan Africa: Protocol for a diagnostic test accuracy systematic review
AUTHORS	Vengesai, Arthur; Naicker, Thajasvarie; Kasambala, Maritha; Midzi, Herald; Mduluzza-Jokonya, Tariro; Rusakaniko, Simbarashe; Mduluzza, Takafira

VERSION 1 – REVIEW

REVIEWER	Barbara Castro-Pimentel Figueiredo Universidade Federal da Bahia
REVIEW RETURNED	27-Oct-2020

GENERAL COMMENTS	In this study protocol, Vengesai et al. describe their plans for improving the diagnosis of neglected tropical diseases. This group will perform a meta-analysis using hierarchical models to assess the accuracy of peptide arrays previously published. They selected reliable databases and carefully described all the analyses they will further execute. The manuscript is interesting and the study will generate original data if conducted as described in this study protocol. I believe the methodology described in the present protocol study falls within the scope of BMJ and significantly contributes to the field of NTD diagnosis. This manuscript is written in a high quality English language and I could not find major mistakes. So, my recommendation is for the acceptance of the present protocol study.
---

REVIEWER	Hunduma Dinka Adama Science and Technology University, Applied Biology
REVIEW RETURNED	01-Dec-2020

GENERAL COMMENTS	Is not up to the standard - the title of the manuscript and it's content do not much.
---

REVIEWER	Christian Bottomley London School of Hygiene and Tropical Medicine
REVIEW RETURNED	03-Mar-2021

GENERAL COMMENTS	This manuscript describes the protocol for a review of microarray diagnostics for NTDs. The rationale for the review is clearly explained and the proposed methods are sound. I have only a few minor comments. p.3/4 The authors could also mention here that skin snips have traditionally been used to diagnose onchocerciasis. p.5 l.137 Test performance is defined by sensitivity and specificity alone. PPV is a function of prevalence (in addition to sensitivity and
---

	specificity) and is therefore not a test characteristic. p.5 The Expanded Special Project for Elimination of Neglected Tropical Diseases (ESPEN) is targeting onchocerciasis, but this is not one of the diseases included in the review. Why? There are a few typos in the manuscript: E.g. p.6 l. 93 : “or The formal ether concentration technique”. I don’t think the “t” in “The” should be capitalised here. Box 1 Is the sentence “Wuchereria bancrofti is responsible for 90% of the case Treatment is through drugs or surgery” meant to be two sentences? (“Wuchereria bancrofti is responsible for 90% of cases” and “Treatment is through drugs or surgery”.) p.14 l. 307 paragraph should finish with a full stop, not a comma.
--	---

VERSION 1 – AUTHOR RESPONSE

Responses to reviewer comments

Reviewer: 1

Dr. Barbara Castro-Pimentel Figueiredo, Universidade Federal da Bahia

Comments to the Author:

In this study protocol, Vengesai et al. describe their plans for improving the diagnosis of neglected tropical diseases. This group will perform a meta-analysis using hierarchical models to assess the accuracy of peptide arrays previously published. They selected reliable data bases and carefully described all the analyses they will further execute. The manuscript is interesting and the study will generate original data if conducted as described in this study protocol. I believe the methodology described in the present protocol study falls within the scope of BMJ and significantly contributes to the field of NTD diagnosis. This manuscript is written in a high quality English language and I could not find major mistakes. So, my recommendation is for the acceptance of the present protocol study.

Response: Thank you for the encouraging comments. Yes, we saw it befitting to run an analysis on the diagnostic methods potential for use following the success of peptide arrays applications. We hope that our studies would generate enough proof for the peptide arrays use.

Reviewer: 2

Dr. Hunduma Dinka, Adama Science and Technology University

Comments to the Author:

Is not up to the standard - the title of the manuscript and it's content do not much.

Response: The title was made to cover a broad application of peptide microarray in the NTDs diagnostic field. Thus enabled us to collect enough of work covered in the area as will be shown by the final analysis. The concerns raised by the reviewer will be addressed by the final output of the systematic review that match more with the title of the manuscript.

We also appreciate the observation and the comments made by the reviewer such that we have gone through the manuscript making the necessary corrections. We apologize for the spelling mistakes we made in the previous submission. We have extensively run the spell check and addressed the concerns. We hope the revision done on the manuscript makes it clearer for easy understanding.

Reviewer: 3

Dr. Christian Bottomley, London School of Hygiene and Tropical Medicine

Comments to the Author:

1. This manuscript describes the protocol for a review of microarray diagnostics for NTDs. The rationale for the review is clearly explained and the proposed methods are sound. I have only a few minor comments.

p.3/4 The authors could also mention here that skin snips have traditionally been used to diagnose onchocerciasis.

p.5 l.137 Test performance is defined by sensitivity and specificity alone. PPV is a function of prevalence (in addition to sensitivity and specificity) and is therefore not a test characteristic.

p.5 The Expanded Special Project for Elimination of Neglected Tropical Diseases (ESPEN) is targeting onchocerciasis, but this is not one of the diseases included in the review. Why?.

Response: An addition was made to include onchocerciasis as part of the study. Below are the sections and lines where onchocerciasis was included.

Abstract.

Line: 28-29

Introduction

Lines: 96-99

Lines: 108-111

Line: 114

Line: 142

Methodology

Line: 179 (table 5)

Line: 215

Line: 220 (table 6)

2. There are a few typos in the manuscript:

E.g.

p.6 l. 93 : “or The formal ether concentration technique”. I don’t think the “t” in “The” should be capitalised here.

Response: Correction made

Comment;

Line: 92

Box 1 Is the sentence “*Wuchereria bancrofti* is responsible for 90% of the case, Treatment is through drugs or surgery” meant to be two sentences? “*Wuchereria bancrofti* is responsible for 90% of cases” and “Treatment is through drugs or surgery”.)

Response: No correction made because Box 1 was deleted from the manuscript as per specifications of the BMJ journal that does not require Box texts.

Comment:

p.14 l. 307 paragraph should finish with a full stop, not a comma.

Response: The comma was replaced with a full stop.

Comment:

Line: 308

Response: The authors also took an additional step to go over the whole manuscript correcting typos and grammar.